# Soil fauna-microbial interactions shifts fungal and bacterial communities under a contamination disturbance

**Sara Correa-Garcia[1,2], Vincenzo Corelli[1], Julien Tremblay[3], Jessica Ann Dozois[1], Eugenie Mukula[1], Armand Séguin**[2], **Etienne Yergeau**[1] *

1 Centre Armand-Frappier Santé Biotechnologie, Institut national de la recherche scientifique, Université du Québec, Laval, QC, Canada, 2 Laurentian Forestry Center, Natural Resources Canada, Québec City, QC, Canada, 3 Energy, Mining and Environment, National Research Council Canada, Montréal, QC, Canada

* etienne.yergeau@inrs.ca

**Data Availability Statement:** Raw sequencing reads were deposited in the NCBI SRA under BioProject accession PRJNA700608. The data matrices used for the statistical analysis can be

## Abstract

The aim of this study was to determine whether the soil faunal-microbial interaction complexity (SFMIC) is a significant factor influencing the soil microbial communities and the willow growth in the context of PAH contamination. The SFMIC treatment had eight levels: just the microbial community, or the microbial community with nematodes, springtails, earthworms and all the possible combinations. SFMIC affected the height and biomass of willows after eight weeks or growth. SFMIC affected the structure and the composition of the bacterial, archaeal and fungal communities, with significant effects of SFMIC on the relative abundance of fungal genera such as *Sphaerosporella*, a known willow symbiont during phytoremediation, and bacterial phyla such as *Actinobacteriota*, containing many polycyclic aromatic hydrocarbons (PAH) degraders. These SFMIC effects on microbial communities were not clearly reflected in the community structure and abundance of PAH degraders, even though some degraders related to *Actinobacteriota* and the diversity of Gram-negative degraders were affected by the SFMIC treatments. Over 95% of PAH was degraded in all pots at the end of the experiment. Overall, our results suggest that, under our experimental conditions, SFMIC changes willow phytoremediation outcomes.

## Introduction

Polycyclic aromatic hydrocarbons (PAH) are organic compounds originating from the incomplete combustion of organic matter. They are considered as contaminants in the environment [1, 2], especially in soils [3], due to their detrimental effects on ecosystems [4]. Phytoremediation has demonstrated a significant potential to tackle PAH contamination [5–9]. Rhizoremediation, the process where microorganisms are stimulated to degrade contaminants in the rhizosphere environment by root exudates [10], is one of the main phytoremediation approaches used to degrade PAH.

However, despite extensive research on PAH phytoremediation, the complexity of soil hinders our understanding of it. Soil complexity arises from a combination of factors, but one of

found in https://doi.org/10.5281/zenodo.8299107The scripts used for the statistical analysis can be found in https://github.com/le-labo-yergeau/SFMIC_amplicon_analyses.

**Funding:** This work was supported by the Natural Sciences and Engineering Research Council of Canada (Discovery grant RGPIN-2014-05274 and strategic grant for projects STPGP 494702) to E.Y. This research was enabled in part by support provided by Compute Canada. The funders had no role in study design, data collection and analysis, decision to publish, or preparation of the manuscript.

**Competing interests:** The authors have declared that no competing interests exist.

the most overlooked in the context of phytoremediation is the interaction of soil fauna with soil microbes. And yet, soil fauna can directly and indirectly regulate soil bacterial and fungal communities. Nematodes grazing of bacterial or fungal cells directly shifts the microbial community [11], whereas collembolans feeding on plant secondary roots can indirectly stimulate the growth of specific microbes in the rhizosphere [12–14]. Similarly, earthworms change soil physicochemical characteristics by degrading organic matter, turning over nutrients, and oxygenating soil, thereby indirectly shifting the microbial community [15–18].

A common PAH used in phytoremediation studies is phenanthrene (PHE). Phenanthrene (PHE) is a PAH formed by three fused benzenic rings. It has been widely used as a model molecule, alone or with other PAHs in degradation experiments [19–22]. Many soil microorganisms metabolize PHE and other PAHs aerobically [23] using a ring hydroxylating dioxygenase multicomponent enzyme (PAH-RHD) [24]. The genes coding for the alpha subunit of the PAH-RHD form a monophyletic group [25] commonly used as biomarkers to monitor bacterial degraders in PAH contaminated environments [26]. Many bacteria containing PAH-RHD genes are found in the root systems of *Salicaceae* trees [22, 27–30], which could explain why these trees are efficient to remediate hydrocarbons [30–34], including PHE [22, 35]. The outcome of phytoremediation remains, however, often unpredictable and variable [10, 36], partially due to the initial composition and diversity of the soil microbial community [27, 37, 38], and soil physicochemical characteristics [22], but also potentially to unmeasured differences in soil fauna.

We think that collembolans, nematodes, earthworms and the rest of the soil fauna may have unexplored and interactive roles in the rhizoremediation of PAH [39, 40], through direct and indirect changes in soil microbial communities. Only a few studies have, however, explored this, the majority of which focusing on a single species or ecological group [41, 42]. Many animals survive in contaminated soils [41, 43–45], but we don't know how the complexity of the soil fauna affects microbial communities, plant performance and contaminant fate during soil phytoremediation. We hypothesized that, because it will increase microbial and nutrient turnover, a more complex fauna during PHE phytoremediation will result in 1) shifts in microbial communities and, consequently, 2) larger plants and 3) enhanced PHE degradation rates. We assessed the response of microbial communities and willow trees to an increasingly complex soil fauna treatment during PHE rhizoremediation.

## Material and methods

### Soil and biological material

The soil was acquired from Savaria Matériaux paysagers Ltée (Laval, Canada) and it was gamma irradiated at a dose ranging from 12.4 to 24.8 kGy by Nordion (Laval, Canada) to significantly reduce the community of soil dwelling invertebrates whilst preserving a significant fraction of the microbial diversity and abundance [46]. Funnel extractions with heat irradiation confirmed the effectiveness of the gamma irradiation, with no arthropods recovered after 48h of extraction. A wet extraction to recover nematodes was not performed since previous experiments report recovering 100% empty carcases of nematodes several weeks after soil irradiation with doses as low as 3 kGy [47, 48]. The control potting mixture was prepared by homogenizing perlite and the irradiated soil in a 1:2 (v:v) proportion. For the contaminated soil, after gamma irradiation, the soil was spiked with 100 mg·kg$^{-1}$ dry soil of phenanthrene according to the following protocol. Batches of 1 kg of dried, 2 mm sieved soil were spiked with 1 g of phenanthrene (Sigma Aldrich) diluted in 100 mL of acetone. Batches of control soil were spiked with acetone alone. The spiked soil was left in a chemical hood for 48h until the acetone was completely evaporated. Then, the spiked 1 kg batches (contaminated or control)

were incorporated into 9 kg of irradiated soil in a cement mixer. Next, perlite was mixed into the soil in a proportion of 1:2 (v:v) until the potting mixture reached homogeneity.

Willow cuttings (*Salix purpurea* cv. FishCreek, 40 cm long) were acquired from Agro Énergie (Saint-Roch-de-l'Achigan, Quebec, Canada). Prior to planting, cuttings were pre-soaked by submerging in tap water at room temperature for one week. Cuttings showing signs of disruption of dormancy (greener buds, incipient root tips) were selected for the experiment. Lab-reared earthworms from the species *Aporrectodea caliginosa*, *Caenorhabditis sp.* nematodes and *Folsomia sp.* springtails were used to create the SFMIC treatment, along with *Tomocerus sp.* springtails purchased at Magazoo (Montreal, Canada). The procedures to collect the animals and the laboratory rearing conditions are detailed in the Supplemental material.

## Experimental design

A full factorial experiment consisting of three factors was implemented: contamination, food web complexity and plant compartment. The contamination treatment consisted of two levels: soils contaminated with 100 mg·kg$^{-1}$ dry soil of phenanthrene (PHE) and an uncontaminated control soil (CTRL). The soil was gamma-irradiated to allow for the control of the food web complexity treatment. The food web complexity treatments consisted of eight levels: bacteria and fungi as control (BF, the irradiated soil only), the irradiated soil (BF) plus springtails (C), nematodes (N) or earthworms (E), and BF plus all the possible two and three animal combinations (CE, CN, EN, CEN). Microbial communities and phenanthrene concentrations were evaluated in two soil compartments: bulk (Bulk) and rhizosphere (Rhizo) soils. The two first factors (contamination and food web complexity) resulted in 16 soil treatments that were used for the pot experiment. The pots were arranged in six experimental blocks, wherein the 16 treatments were randomly distributed, for a total of 96 pots that were placed outside at the Centre Armand Frappier Santé Biotechnologie (INRS, Laval, Canada, 45.541393˚N, -73.716980˚W). One cutting was planted per six-liter pot containing approximately 4 kg of potting mixture on the 7th and 8th of August 2018. Animals were added immediately after the cuttings were planted. Five adult earthworm specimens presenting a fully developed clitellum were deposited at the surface of the corresponding pots. A water solution of nematodes was prepared and 5 mL of it, containing approximately 5,000 individuals of all developmental stages, were inoculated at the soil surface. A mixture of *Tomocerus* and *Folsomia* colonies of approximately 200 individuals was inoculated 5 cm under the soil surface. Pots were then covered with coconut fiber to prevent animals from escaping and to reduce phenanthrene evaporation and photooxidation. Then, the pots were connected to an automated drip irrigation system and received 400 mL of water daily.

## Sampling and plant trait measurements

Samples were collected between the 1st and 4th of October 2018, after approximately 8 weeks of plant growth. The duration of the growth stage was based on the degradation rate observed in a previous pot experiment with poplars and phenanthrene [22]. Plant height and the number of shoots were measured prior to clipping aboveground shoot biomass (excluding the original cutting). Afterwards, the cuttings with the attached roots were taken out from the pots and rhizosphere and bulk soil samples were collected for microbial community analysis and phenanthrene concentration measurements as described in [22], resulting in 192 soil samples. Soil samples were kept at 4˚C until transportation to the lab (around 2 h) where they were placed at -20˚C. For all plants harvested, aboveground willow biomass was weighed fresh, oven-dried for 24 h at 60˚C and weighed again to recover dried biomass and aboveground water content values. Root biomass was collected by sieving all the soil from each pot through a 2 mm mesh

to ensure recovery of the maximum amount of root biomass. Roots collected were subsequently cleaned with water from perlite and adhered soil, then dried and weighed. The objective of our study was not to determine the effects of PHE on the survival and fitness of soil invertebrates; therefore no animal extraction was performed. However, the presence of variable numbers of surviving E and C was visually confirmed on the pots receiving these treatments at the end of the experiment.

## Phenanthrene quantification

Phenanthrene concentration was measured in the rhizosphere and bulk soil samples taken at the end of experiment. Two phenanthrene measurements were taken per biological replicate. The phenanthrene extraction protocol consisted on an in-house method developed in the lab and described in [22]. Briefly, 4 g of frozen soil were mixed with 900 μL of ethyl acetate, 3 mL of distilled water and 10 ppm of phenanthrene-$d_{10}$ in 100 μL of ethyl acetate used as internal standard. Then, the samples were placed in an ultrasonic bath at 60 kHz for 15 min. Next, the samples were shaken overnight at 300 rpm at room temperature. Afterward, the samples were centrifuged at 270 g for 10 min and the organic phase was recovered. The organic phase was further purified by centrifugation at $5,000 \times g$ for 1 min to yield the final phenanthrene extracts to be used in Gas Chromatography -Mass Spectrometry analysis (GC-MS).

Phenanthrene extracts were analyzed using a Trace GC Ultra system (Thermo Scientific) with a 30 x 0.25 mm (0.25 μm thickness) DB-5 MS capillary column (Agilent J & W capillary GC) and coupled to a Polaris Q benchtop Ion Trap Mass Spectrometer. The injector temperature was set at 250˚C and the analyzer at 350˚C. The GC-MS program consisted in 2 min hold at 70˚C, increasing temperature to 310˚C at 30˚C $min^{-1}$ followed by 6 min hold at 310˚C. Helium was used as carrier gas at a flow rate of 0,3 mL/min. The injection volume was 3 μL. MS scan range was set at 70–600 m/z.

Standard calibration curves for serial concentrations ranging from 1 ppm to 100 ppm were calculated by plotting the peak areas against the concentration of reference. The final concentration of phenanthrene is expressed as mg $kg^{-1}$.

## Microbial community composition: DNA isolation and library preparation

The DNA isolation steps and the library preparation follow the steps described in-depth in [22]. Briefly, 250 mg per sample of bulk or rhizosphere soil were homogenized in a FastPrep®-24 (MP Biomedicals). Then, DNA was extracted using the DNeasy® Powersoil® kit (Qiagen) following the protocol. The V3-V4 region of the 16S rRNA gene, the internal transcribed spacer region (ITS), and the PAH-RHDα gene clusters for Gram negative (GN) and Gram positive (GP) bacteria were amplified using the primer sets 515F – 806R [49], ITS1F – 58A2R [50], 610F – 916R and 641F – 933R [26], respectively. All microbial regions were amplified in 25 μL volumes containing 10–20 ng of DNA template as in [22]. The polymerase chain reaction (PCR) conditions for bacteria and archaea were as follows: initial denaturation at 95˚C for 5 min; 35 cycles at 95˚C for 30 s, 55˚C for 30 s, 72˚C for 1 min, and a final elongation phase at 72˚C for 5 minutes. For fungi, the annealing temperature was set at 59˚C instead of 55˚C, and the number of cycles at 30 instead of 35. PAH-RHDα genes had annealing temperatures set at 57˚C and 54˚C for GN and GP primer sets, respectively and a total of 30 cycles. PCR products were cleaned following the Illumina's protocol "16S Metagenomic Sequencing Library preparation" guide (Part #15044223 Rev. B). Then, clean PCR products were tagged using 400 nM of each Nextera XT index primers under the thermal cycling conditions: 95˚C initial denaturation phase for 5 min, followed by 8 cycles consisting of 95˚C denaturation for 30 s, annealing at 55˚C for 30 s, elongation at 68˚C for 30 s, and a final elongation

phase at 68˚C during 5 min. These libraries were sequenced on an Illumina Miseq sequencer at the Centre d'expertise et de services Génome Québec (Montréal, QC, Canada).

## Real-time PCR quantification of PAH degrading genes

The real-time quantitative PCR (qPCR) was conducted on a Stratagene Mx3005P qPCR system (Agilent Technologies). The qPCR reactions were performed using the primers designed by [26] as in [22]. Briefly, the denaturation step at 95˚C for 5 min was followed by 40 cycles of denaturation at 95˚C for 30 s, annealing for 35 s at either 57˚C (GN) or 54˚C (GP) and an elongation step at 72˚C for 75 s, after which the SYBR Green signal intensity was measured. A melting curve analysis was performed where signal intensity was measured at 0.5˚C increments every 5 s from 51 to 95˚C. Standards were made from 10-fold dilutions of linearized plasmid containing the gene fragment of interest, cloned from soil DNA [51].

At the end of the run, the cycle threshold (Ct) values were evaluated, and the gene copy numbers were calculated from the standard curve based on the Ct values. The efficacy of the qPCRs ranged from 49.6% ($R^2$ = 0.994) to 85.0% ($R^2$ = 0.996).

## Bioinformatic analyses

Sequences were analyzed using AmpliconTagger [52]. Briefly, raw reads were scanned for sequencing adapters and PhiX spike-in sequences. Remaining reads with an average quality (Phred) score lower than 20 were discarded. The rest of the sequences were processed for generating Amplicon Sequence Variants (ASV; DADA2 v1.12.1; [53]). Chimeras were removed with DADA2's internal removeBimeraDeNovo(method ="consensus") method followed by UCHIME reference [54]. Only ASVs with an abundance across all samples higher than 5 were retained. A global read count summary throughout the pipeline is provided for all datasets (S1 Table). ASVs were assigned a taxonomic lineage with the Ribosomal Database Project (RDP) classifier [55] using an in-house training set based on the complete Silva release 138 database [56] supplemented with eukaryotic sequences from the Silva database and a customized set of mitochondria, plasmid and bacterial 16S sequences. For ITS ASVs, a training set containing the Unite DB to classify sequences (sh_general_release_s_04.02.2020 version) was applied. The final lineages were reconstructed using the taxonomic depths having a score ≥ 0.5 (from a 0 to 1 range assigned by the RDP classifier). Taxonomic lineages were combined with the cluster abundance matrix obtained above to generate a raw ASV table, from which a bacterial/fungal organisms ASV table was generated.

PAH-RHDα GP and GN amplicon sequencing libraries were processed as described above up to the quality filtering step and remaining sequences were processed to generate ASVs (DADA2 v1.12.1) [53]. Custom RDP classifier training sets were generated for both PAH-RHDα GP and GN amplicon data types as follow. Each ASV was compared against the National Center for Biotechnology Information (NCBI) nt database (downloaded on February 21, 2020) with–max_target_seqs set to 20 to find regions of local similarity between sequences (BLASTn). Blast output was filtered to keep hits using the following thresholds: an e-value < = 1e-20, alignment length of at least 100 bp and alignment percentage of at least 60%. Taxonomic lineages of each filtered blast hit were fetched from the NCBI taxonomy database. RDP training sets were generated as previously described (https://github.com/jtremblay/RDP-training-sets).

## Statistical analyses

Statistical analyses were performed in R (v.3.5.0) [57]. Linear regression assessed the effect of residual phenanthrene concentrations and plant biomass on the observed number of ASV, Shannon H' index, and PAH-RHD Gram Positive and Gram Negative degradation genes. The

differences in plant traits, qPCR quantifications, phenanthrene concentrations and diversity indices (Shannon H' and observed number of ASV, calculated with the otuSummary package [58]) were assessed with analysis of variance (ANOVA) followed by *post hoc* honestly significant difference tests (Tukey HSD). Kruskal-Wallis analysis of variance coupled with the Dunn test as *post hoc* when the assumptions of normality and homoscedasticity were not met.

The differences in the community structure were visually assessed using principal coordinate analyses (PCoA) with normalized ASV tables and the Bray-Curtis dissimilarity index calculated with the vegan package [59]. The effect of the treatments was tested using permutational multivariate analysis of variance (PERMANOVA) analyses with 999 permutations using the adonis2 function. The effect of plant biomass and phenanthrene concentrations on the microbial community structure was evaluated with distance-based redundancy analysis (db-RDA). All graphs were created with the ggplot2 package [60].

## Results

### Plant biomass and phenanthrene concentration

The average fresh biomass of willow shoots was 38.73 g across treatments. The linear model showed that the contamination treatment led to a reduction of 8.29 g of fresh shoot biomass compared to controls (ANOVA: $F = 15.144$, $p < 0.001$; Table 1 and S2 Table). Conversely, the shoot fresh biomass responded positively to the soil fauna-microbial interactions complexity (SFMIC) treatment (ANOVA: $F = 3.003$, $p < 0.01$). Specifically, pots with the treatments CEN, E, CE and EN exhibited a significantly increased in fresh shoot biomass by 15.56 g, 10.20 g, 8.77 g, and 9.99 g, respectively, compared to non-inoculated pots (BF; S2 Table). Generally, the willows with the highest shoot fresh biomass were found in CTRL soils, with the notable

**Table 1. Mean and standard deviation for willow morphological traits.** The letters at the right side of std denote significantly different groups as stated by the Tukey post hoc test.

| SFMIC | Cont. | Fresh aerial biomass (g) | | | | Total dry biomass (g) | | | | Height (cm) | | | |
|---|---|---|---|---|---|---|---|---|---|---|---|---|---|
| | | mean | | std | | mean | | std | | mean | | std | |
| BF | CTRL | 39.771 | ± | 9.989 | ab | 16.379 | ± | 4.852 | ab | 96.5 | ± | 7.423 | ab |
| | PHE | 24.166 | ± | 4.149 | b | 9.67 | ± | 2.177 | b | 76.17 | ± | 5.307 | b |
| CEN | CTRL | 50.542 | ± | 17.757 | a | 19.062 | ± | 7.321 | a | 96.83 | ± | 10.534 | ab |
| | PHE | 44.51 | ± | 13.843 | ab | 16.066 | ± | 4.336 | ab | 90.33 | ± | 16.464 | ab |
| C | CTRL | 40.752 | ± | 6.718 | ab | 16.54 | ± | 3.305 | ab | 91.75 | ± | 8.507 | ab |
| | PHE | 31.013 | ± | 5.053 | ab | 12.041 | ± | 2.165 | ab | 81.67 | ± | 13.397 | ab |
| N | CTRL | 37.773 | ± | 5.857 | ab | 14.305 | ± | 3.144 | ab | 89.5 | ± | 4.806 | ab |
| | PHE | 29.856 | ± | 16.586 | ab | 11.4 | ± | 6.149 | ab | 81.67 | ± | 23.036 | ab |
| E | CTRL | 44.483 | ± | 7.462 | ab | 17.028 | ± | 2.986 | ab | 91.75 | ± | 5.707 | ab |
| | PHE | 39.862 | ± | 8.949 | ab | 14.949 | ± | 3.523 | ab | 92.42 | ± | 6.829 | ab |
| CE | CTRL | 41.609 | ± | 13.081 | ab | 16.373 | ± | 6.874 | ab | 91.92 | ± | 7.697 | ab |
| | PHE | 39.886 | ± | 5.249 | ab | 15.512 | ± | 3.207 | ab | 87.67 | ± | 12.242 | ab |
| CN | CTRL | 40.651 | ± | 13.615 | ab | 15.76 | ± | 5.488 | ab | 95.08 | ± | 8.857 | ab |
| | PHE | 30.841 | ± | 8.299 | ab | 12.343 | ± | 3.167 | ab | 87 | ± | 7.239 | ab |
| EN | CTRL | 47.372 | ± | 13.038 | a | 17.83 | ± | 4.93 | ab | 99.17 | ± | 11.839 | a |
| | PHE | 36.539 | ± | 7.378 | ab | 13.324 | ± | 2.166 | ab | 87.5 | ± | 7.342 | ab |

SFMIC: Soil fauna-microbial interactions complexity; Cont: contamination; CTRL: control pots; PHE: phenanthrene contaminated pots. BF: naturally present Fungi and Bacteria. CEN: BF and collembola, earthworms and nematodes; C: BF and collembola; N: BF and nematodes; E: BF and earthworms; CE: BF and collembola and earthworms; CN: BF and collembola and nematodes and EN: BF and earthworms and nematodes.

exception of CEN PHE willows, which produced the third highest fresh shoot biomass of all treatments (Table 1).

The average dry biomass of willows across treatments was 14.91 g (shoots and roots combined). The contamination treatment resulted in an average reduction of dry biomass of 3.50 g compared to CTRL willows ($F$ = 15.703, $p$ < 0.001; Table 1 and S2 Table). In this case, the SFMIC had a limited impact on the dry biomass of willows ($F$ = 1.709, $p$ = 0.117), and only the CEN willows experienced an average increase of 4.54 g in dry biomass compared to BF pots (S2 Table). Still, the PHE CEN willows produced similar amounts of biomass to willows growing in CTRL soil, and all willows growing in PHE soils with earthworms had higher total biomass than the willows growing in PHE soils without earthworms (Table 1). Only the total biomass of the CRTL CEN and the PHE BF treatments were significantly different in *post-hoc* tests (S2 Table).

The average height of the willows was 89.81 cm. Again, on average, contamination reduced the height by 8.51 cm compared to CTRL ($F$ = 14.828, $p$ < 0.001). Although the overall effect of the SFMIC was not significant (F = 1.066, $p$ = 0.39), willows growing in PHE soil in combination with E were taller than those in the BF treatment (Table 1).

At the end of the experiment, the rhizosphere compartment presented 0.24 mg kg$^{-1}$ more residual phenanthrene than the bulk compartment ($F$ = 6.27, $p$ < 0.05). However, this difference was not biologically significant since the highest retrieved phenanthrene value was of 3.99 mg kg$^{-1}$, and the lowest 0.89 mg kg$^{-1}$ (Fig 1). Overall degradation rates ranged from 96 to 99% of the 100 mg kg$^{-1}$ phenanthrene applied, both in the bulk and rhizosphere soils. The SFMIC did not have an impact on phenanthrene degradation ($F$ = 1.508, $p$ = 0.176).

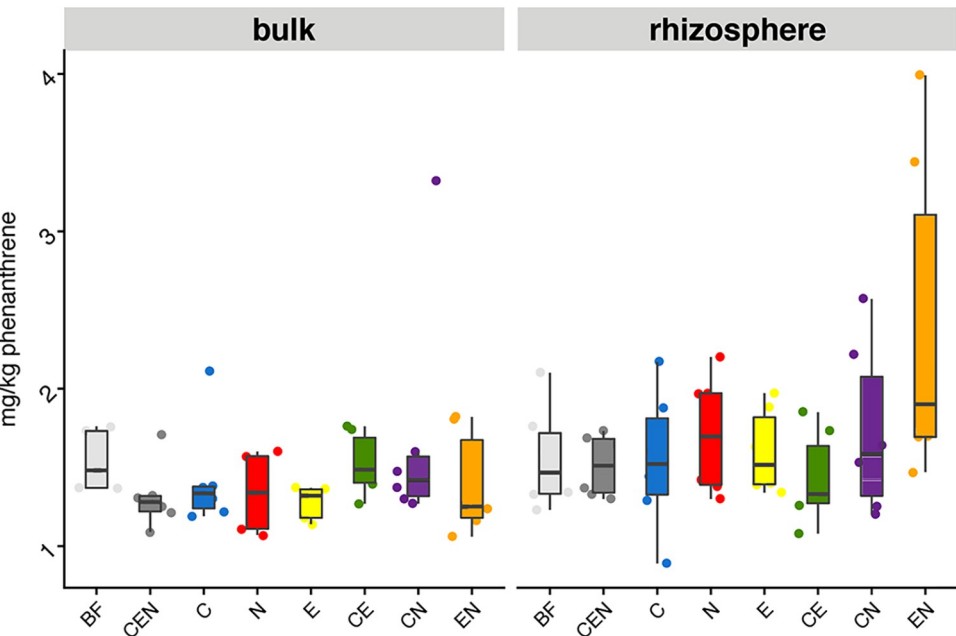

**Fig 1. Quantification of phenanthrene (as mg kg$^{-1}$ soil) after 8 weeks of willow growth in bulk and rhizosphere soil compartments for all soil fauna-microbial interactions complexity (SFMIC) treatments.** No significant differences were found (n = 6). Legend: SFMIC levels: BF = only naturally present *Bacteria* and *Fungi*; CEN = BF plus added collembola, nematodes and earthworms; C = BF and added collembola; N = BF and added nematodes; E = BF and added earthworms; CE = BF and added collembola and earthworms; CN = BF and added collembola and nematodes; EN = BF and added nematodes and earthworms.

## Microbial community diversity

The median values for the Shannon H' diversity index and the number of observed ASV were calculated for the four sets of gene biomarkers (S3 Table). Linear models and the ANOVA results were calculated for both diversity index and for all biomarkers (S4 and S5 Tables).

The contamination, the compartment and the SFMIC treatments significantly affected the fungal Shannon H' diversity. In average, the fungal Shannon H' diversity decreased by 0.617 points in PHE samples. In the rhizosphere, the index was 0.35 higher than in bulk soils. Compared to BF samples, the average fungal diversity index increased by 0.66 and 0.53 in the N and CEN samples, respectively. Also, the linear regression model showed that dry biomass did not have an impact on fungal Shannon diversity (F = 0.534, $p$ = 0.466), but in accordance with the contamination effect, 1 mg kg$^{-1}$ of residual phenanthrene caused in average a reduction of 0.18 points compared to controls (F = 8.835, $p <$ 0.01).

The bacterial and archaeal Shannon H' diversity was significantly influenced by contamination and the soil compartment. PHE pots and rhizosphere soils were more diverse as compared to CTRL and bulk compartments (0.232 and 0.186 points, respectively). Interestingly, the rhizosphere of PHE CN (5.58), CEN (5.48), CE (5.45) and EN (5.44) had the highest Shannon H' diversity (other values ranging from 5.41 to 4.72). The residual phenanthrene was associated with a small increase of the bacterial Shannon index (0.06 points, F = 8.927, $p <$ 0.001), but the biomass did not cause any significant change (F = 2.716, $p$ = 0.101).

The Shannon H' diversity of Gram-negative bacterial PAH degraders experienced a significant increase due to the contamination (0.30 points more in average compared to CTRL) and compartment (0.25 average increase in the rhizosphere compared to bulk soil), but no change due to the SFMIC treatment. Accordingly, the index increased in average 0.102 points per additional mg kg$^{-1}$ residual phenanthrene (F = 13.846, $p <$ 0.001), but the biomass had no significant effect on the diversity of Gram-negative bacterial PAH degraders (F = 1.343, $p$ = 0.248).

Regarding the Gram-positive bacterial degraders, the Shannon H' index was also significantly higher the rhizosphere soil compartment and in the PHE pots. All the treatments with nematodes (N) had the highest absolute diversity. Also, each additional mg kg$^{-1}$ residual phenanthrene was related to a 0.18 increase in the Shannon index (F = 24.652, $p <$ 0.001), and plant biomass did not cause any effect (F = 0.573, $p$ = 0.45).

## Microbial community structure

The fungal community structure was mainly shaped by contamination ($R^2$ = 5.3%, $p <$ 0.001), and the soil fauna-microbial interactions complexity (SFMIC; $R^2$ = 6.1%, $p$ = 0.007; Fig 2A). The PCoA showed CTRL (left side) and PHE (right side) pots dispersed along the first axis. The second axis allowed for a greater dissemination of CTRL fungal community structures, compared to PHE pots (Fig 2A), along with the separation of pots containing collembolans (C and CN) from the ones containing earthworms and/or nematodes (E, N, CE and EN).

For *Bacteria* and *Archaea*, the PERMANOVA analysis (Fig 2B), showed that the contamination ($R^2$ = 11.0%, $p <$ 0.001), the soil compartment ($R^2$ = 3.4%, $p <$ 0.001) and the SFMIC treatment ($R^2$ = 4.4%, $p$ = 0.002), had all significant main effects on the community structure. In addition, the effect of the contamination was modulated by the soil compartment ($R^2$ = 0.7%, $p$ = 0.034). Most of these effects are visible in the PCoA (Fig 2B), as the communities separate along the first axis in two clear groups, with CTRL soils on the left and PHE soil on the right. These clusters further separate between the rhizosphere and bulk soil samples along the second axis. As for the interaction between the contamination and the soil compartment, the samples within the bulk soil and the rhizosphere of the CTRL treatments were more dispersed

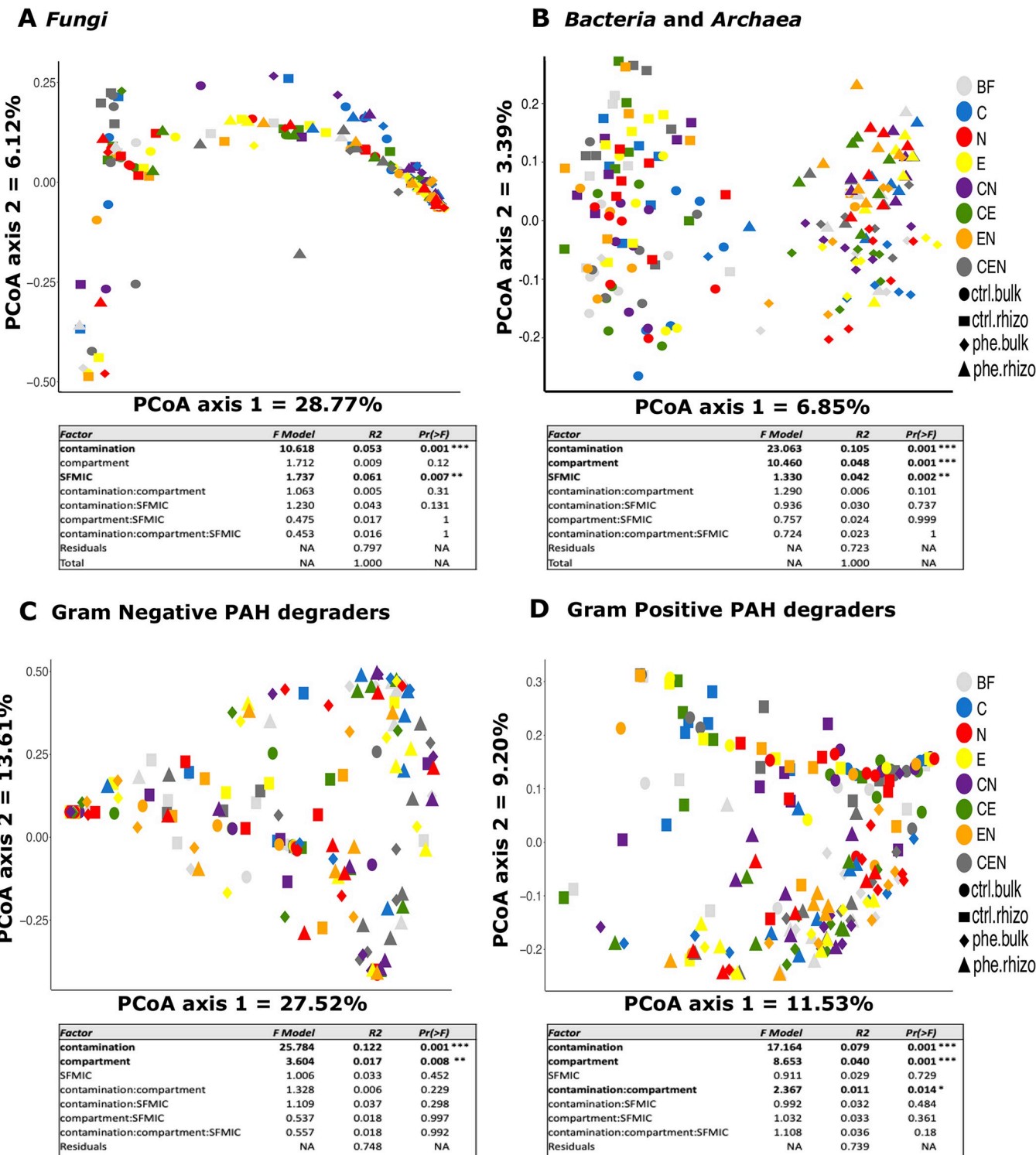

**Fig 2. Principal coordinate analysis (PCoA) based on Bray-Curtis dissimilarity of the relative abundance of ASV showing the effects of contamination, soil food complexity treatment, soil compartment and the respective interactions of these three factors on the community structures. A)** *Fungi* (based on the ITS region); **B)** *Bacteria* and *Archaea* (based on the 16S rRNA gene); **C)** Gram Negative bacterial degraders (based on the PAH-RHDα GN gene) and **D)** Gram Positive bacterial degraders (based on the PAH-RHDα GP gene). Below each PCoA there is the summary table of the permutational multivariate analysis of variance (PERMANOVA) examining the differences in the microbial communities based on the above-mentioned factors. Legend: SFMIC levels: BF = only naturally present *Bacteria* and *Fungi*; CEN = BF plus added collembola, nematodes and earthworms; C = BF and added collembola; N = BF and added

nematodes; E = BF and added earthworms; CE = BF and added collembola and earthworms; CN = BF and added collembola and nematodes; EN = BF and added nematodes and earthworms; con.bul: bulk compartment in control pot; con.rhi: rhizosphere compartment in control pot; phe.bul: bulk compartment in phenanthrene contaminated pot and phe.rhi: rhizosphere compartment in phenanthrene contaminated pot.

than the samples from bulk and rhizosphere of the PHE treatment, respectively. The changes in the community structure caused by the SFMIC were not evident on the first two axis of the ordination and the specific effects are presented in the community composition section of this manuscript.

The PERMANOVA analysis of the PAH-RHDα GN gene dataset showed that contamination was the main factor significantly affecting the community structure ($R^2$ = 12.2%, $p < 0.001$; Fig 2C). As opposed to the ordination of the bacterial and archaeal community structures, the PAH-RHDα GN gene dataset did not cluster into clearly identifiable groups. However, it is possible to detect some separation between PHE (right) and CTRL (center-left).

Finally, the PCoA ordination of the PAH-RHDα GP gene dataset showed a stronger separation between PHE (bottom) and CTRL (top) communities (Fig 2D). Again, the main effect shaping the community structure was the contamination level ($R^2$ = 7.9%, $p < 0.001$), but the compartment ($R^2$ = 4.0%, $p < 0.001$) and its interaction with contamination ($R^2$ = 1.1%, $p < 0.001$; Fig 2D) had also significant effects (Fig 2D).

Additionally, we performed 4 db-RDA to assess the joint effect of contamination, SFMIC, compartment, plant biomass, phenanthrene concentration at the end of the experiment, and Gram-positive and Gram-negative degradation gene copy abundance on the microbial community structure. In general, the proportion of inertia explained by the models was low (21.3%, 16.8%, 18.3% and 15.8% for Bacteria, Fungi, GN degraders and GP degraders, respectively). Like PERMANOVA, contamination and compartment were significant factors explaining the structure of the four communities. SFMIC and biomass also contributed to the community structure observed in Bacteria (F = 1.362, $p < 0.001$ and F = 2.275, $p = 0.002$ for SFMIC and biomass, respectively) and Fungi (F = 1.770, $p = 0.004$, F = 3.171, $p = 0.009$ for SFMIC and biomass, respectively).

## Community composition

After sequence data processing in the bioinformatic pipeline, 5,408 fungal ASVs were recovered and classified into 14 phyla, 41 classes, and 599 genera. We found *Ascomycota* to be the dominant phylum. Among the genera, *Sphaerosporella* (Class *Pezizomycetes*; Fig 3A) vastly dominated most plant pots, along with other 13 genera that had a relative abundance higher than 0.5%. Together, these 14 genera accounted for 67.44 to 98.83% of the total ASVs in each treatment combination. The presence of soil fauna significantly changed the relative abundance of the fungal community at genus level. In particular, the presence of collembolans increased the relative abundance of *Chaetomium* (mean relative abundances of 10.04%, 7.26%, 7.60% and 4.41%) and *Zopfiella*, (4.36%, 8.13%, 8.26% and 0.29%) in C, CE, CEN and CN treatments, respectively. In contrast, neither of these genera had relative abundances higher than 1% in the E, EN, N and BF treatments. The changes in the relative abundance of *Sphaerosporella* could not be attributed to a single animal or a specific combination of animals. Other groups were marginally affected by the SFMIC treatment (Fig 3A and S6 Table).

The presence of contamination had a significant effect on the fungal community composition, reducing the relative abundances of several genera while increasing the relative abundance of *Sphaerosporella* (Fig 3A and S6 Table). The soil compartment also had a significant effect on some genera (Fig 3A and S6 Table).

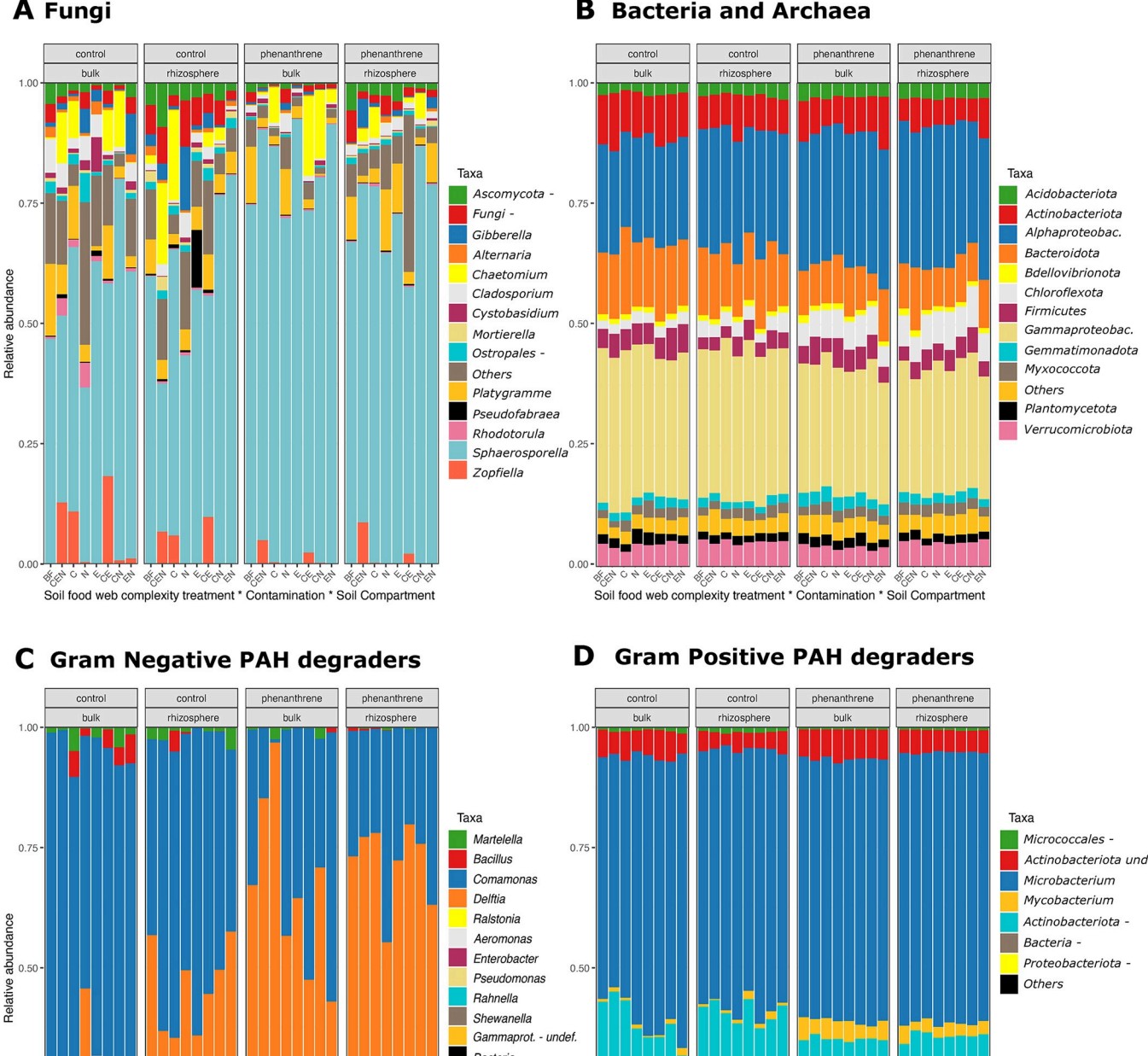

**Fig 3. Microbial community composition based on the relative abundance of ASV.** Values are averaged across treatments. Only taxa with a relative abundance above 0.1% are shown. **A)** Fungal community composition at the genus level. **B)** Bacterial and archaeal community composition at the phylum level. **C)** Gram negative bacterial degraders community composition at the genus level. **D)** Gram positive bacterial degraders community composition at the genus level. Legend: SFMIC levels: BF = only naturally present *Bacteria* and *Fungi*; CEN = BF plus added collembola, nematodes and earthworms; C = BF and added collembola; N = BF and added nematodes; E = BF and added earthworms; CE = BF and added collembola and earthworms; CN = BF and added collembola and nematodes; EN = BF and added nematodes and earthworms. The "-" symbol represents other members of the taxa they accompany (ej. "*Ostropales -*" means "other *Ostropales*").

The bacterial and archaeal communities were represented by 15,154 ASVs belonging to 50 phyla. The majority was classified as *Acidobacteriota, Actinobacteriota, Bacteroidota, Bdellovibrionota, Chloroflexota, Firmicutes, Gemmatimonadota, Myxococcota* (previously classified as an order of *Deltaproteobacteria*), *Planctomycetota, Proteobacteriota* and *Verrucomicrobiota*.

The SFMIC treatment had a marginal effect on the bacterial community at the phylum level, where it modulated the response of *Actinobacteriota* and *Bacteroidota* to the contamination level (S7 Table). In particular, the CEN and the EN treatments increased the relative abundances of *Actinobacteriota* and *Bacteroidota* in the rhizosphere of PHE pots up to levels similar to CTRL, while in bulk soils, the C and the CN treatments reduced the relative abundance of *Bacteroidota* in PHE pots (S1 Fig). Contamination and compartment caused the main effects among *Bacteria* at phylum level relative abundances (S7 Table and 3B Fig).

After quality filtering, 2,818 ASVs were retained for the PAH-RHDα GN genes, classified into 3 phyla (*Deinococcota, Firmicutes, Proteobacteriota*), 6 classes (*Alphaprotebacteria, Betaproteobacteria, Gammaproteobacteria, Bacilli* and *Deinococci* and other *Bacteria*) and 22 genera. Among these, 13 genera had a relative abundance higher than 0.5% (S8 Table). The changes in relative abundance of these genera were not affected by the SFMIC treatment (S8 Table and Fig 3C). Contamination was identified as the main factor contributing to the changes in the relative abundance of these genera, with some also being affected by soil compartment. The contamination led to a reduction in the relative abundance of several genera. However, only the relative abundance of *Delftia* was increased in contaminated pots both in the bulk and the rhizosphere compartments. The soil compartment also affected the relative abundance of both *Comamonas* (more abundant in in bulk compartments) and *Delftia* (more abundant in the rhizosphere).

Following the quality filtering, a total of 6,415 ASVs were retained for PAH-RHDα GP genes, classified into 9 phyla (*Actinobacteriota, Bacteroidota, Chloroflexota, Firmicutes, Planctomycetota, Proteobacteriota, Spirochaetota, Verrucomicrobiota*), 18 classes and 67 genera. Among these, 7 genera had a mean relative abundance exceeding 0.05% (S9 Table and Fig 3D). *Microbacterium* was found to be unaffected by contamination, compartment or SFMIC treatment, accounting for 53–56% of all PAH-RHDα GP ASVs. The contamination was identified as the primary factor responsible for changes in the mean relative abundance of other genera. In contaminated soils, all Actinobacteria genera increased their mean relative abundance in contaminated soils compared to controls. Conversely, other *Micrococcales* and other *Proteobacteriota* experienced a reduction of their relative abundance in contaminated pots compared to controls. The SFMIC treatment had only a slight impact on the mean relative abundance of other *Bacteria*, with a higher relative abundance in the CEN (12.94%) compared to the BF (10.61%) and the other treatments (ranging 7.36% - 9.73%).

## PAH-RHDα GN and GP gene abundance

The SFMIC treatment did not affect the abundance of PAH-RHDα genes. In contrast, the contamination was found to significantly increase the absolute abundance of these genes both in Gram-positive and Gram-negative bacteria (Fig 4). Interestingly, the rhizosphere compartment presented a higher abundance of PAH-RHDα GN genes than the bulk soil. However, the absolute abundance of PAH-RHDα GP genes was only marginally higher in the bulk soil than in the rhizosphere (Fig 4 and S10 Table).

In the regression analysis that evaluated only contaminated pots, the rhizosphere soil showed an average of 463.4 million copies per gram of soil of PAH-RHDα GN genes more than the bulk compartment (F = 119.986, $p < 0.001$). Additionally, an increase of 1 mg/kg increase in residual phenanthrene resulted into an average increase of 103.09 million copies

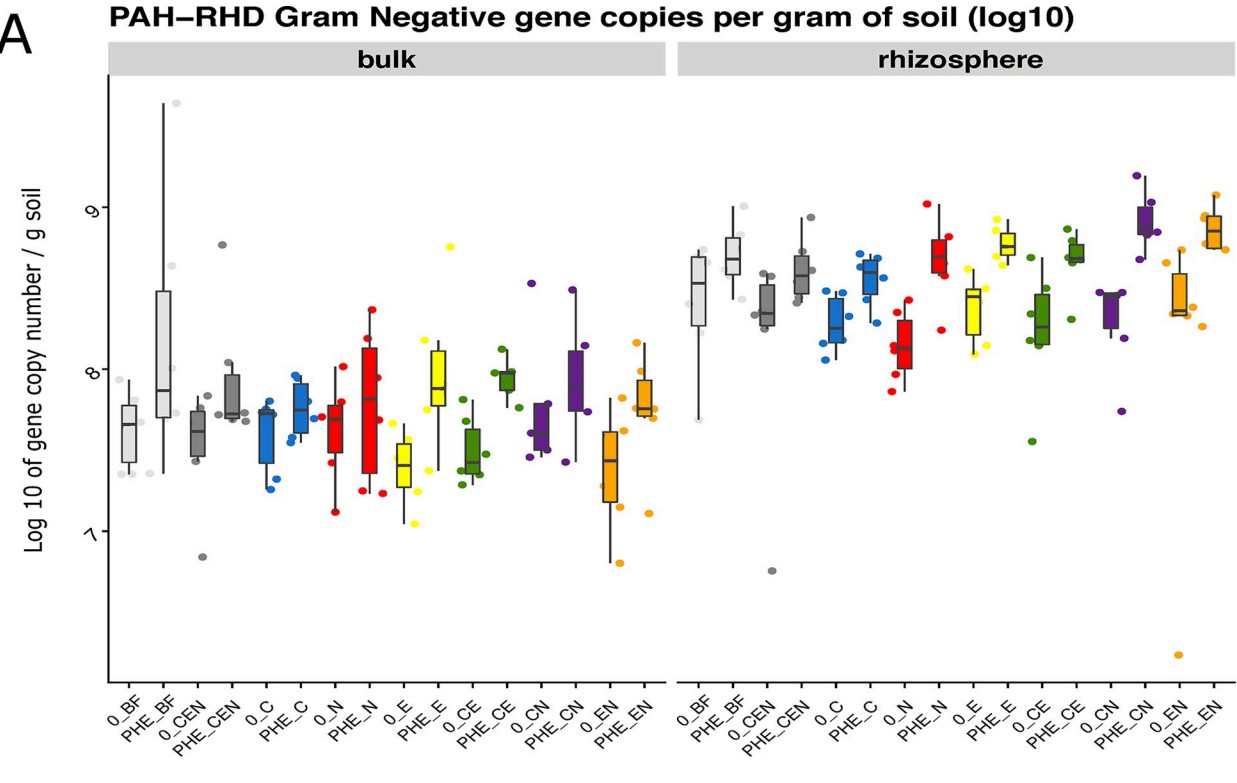

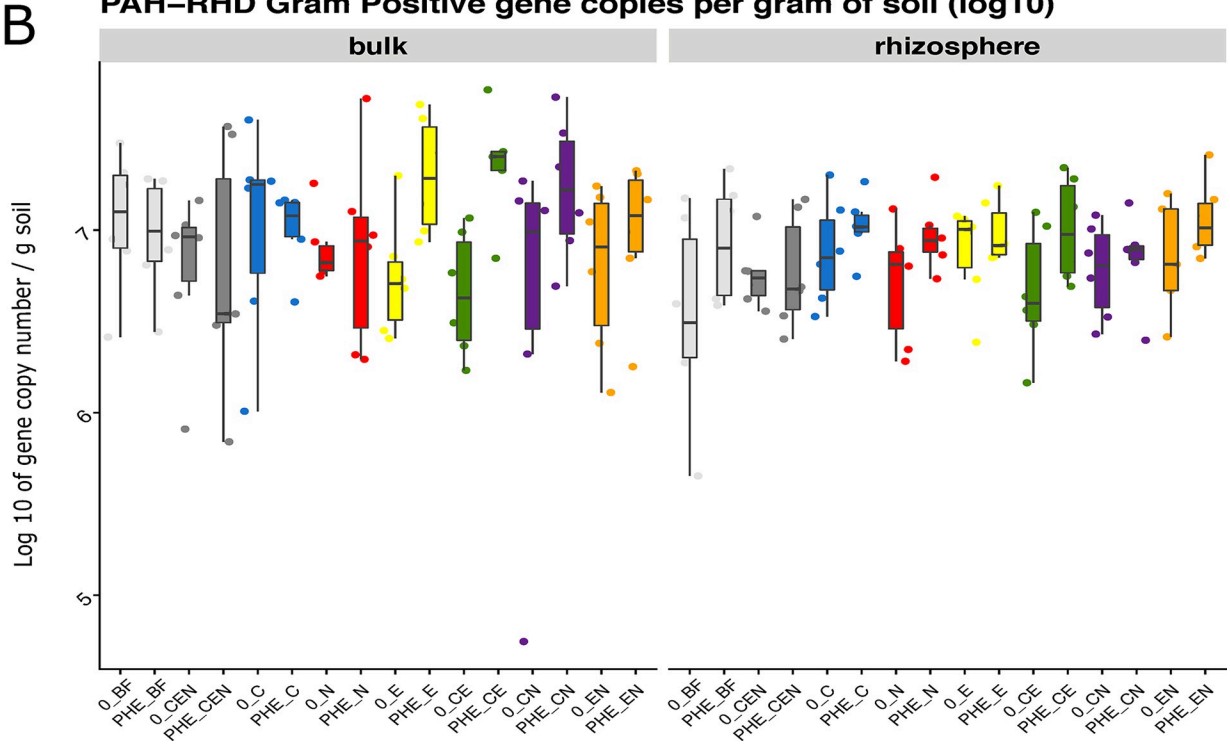

**Fig 4. PAH-RHDα gene copy numbers determined by real-time PCR quantification on DNA. A)** Gram negative bacterial degraders and **B)** Gram positive bacterial degraders. Values are log transformed. Legend: CON_*: control pots; PHE_*: phenanthrene contaminated pots; SFMIC levels: BF = only naturally present *Bacteria* and *Fungi*; CEN = BF plus added collembola, nematodes and earthworms; C = BF and added collembola; N = BF and added nematodes; E = BF and added earthworms; CE = BF and added collembola and earthworms; CN = BF and added collembola and nematodes; EN = BF and added nematodes and earthworms.

per gram of soil of PAH-RHDα GN genes (F = 4.747, $p < 0.05$). In contrast, the rhizosphere compartment presented in average 7.5 million copies of the PAH-RHD GP gene less than the bulk soil (F = 10.090, $p < 0.01$), whereas an increase of 1 gram in biomass caused an increase of 0.725 million copies of the GP gene set (F = 5.955, $p < 0.05$).

## Discussion

Although there is ample evidence that soil fauna influences soil microbial communities [41, 61–63], not many studies have focused on how soil fauna effects on plant and microbes can impact phytoremediation. Since microbes are key players in contaminant degradation, soil fauna could indirectly affect remediation efficiency by altering microbial community composition. In this study, we investigated the effects of soil fauna manipulations (all combinations of the presence/absence of collembolans, nematodes and earthworms) on the microbial communities in bulk soil and willow rhizosphere during phytoremediation. In short, we found that the soil faunal-microbial interactions complexity (SFMIC) affected fungal and bacterial communities, yielded slightly larger trees but did not affect phenanthrene degradation.

Our first hypothesis that the SFMIC treatments would affect the microbial communities was partly confirmed with significant effects on the community structure and composition of Fungi, Bacteria and Archaea. The SFMIC treatments also modulated the effect of the contamination on Bacteria. For instance, in the pots treated with earthworms and nematodes (EN) or collembolans, earthworms and nematodes (CEN) the *Actinobacteriota* and *Bacteroidota* were more abundant in the contaminated rhizosphere as compared to the control rhizosphere–contrasting with the trend observed for the other treatments. Accordingly, the Bacteroidota was shown to be more abundant in the presence of the earthworms *Pontoscolex corethrurus* [64], *Eisenia fetida* [65], *Lumbricus terrestris* and *Aporrectodea caliginosa* [66]. The soil *Actinobacteriota* abundance can also increase after passage through the earthworm gut [65, 67–69]. The trends observed for *Actinobacteriota* were mirrored for the Gram-positive PAH degraders related to this phylum. This specific effect of SFMIC on *Actinobacteriota* could partially explain why earthworms have proven to be useful during PAH remediation [70–73].

As for fungi, we found a significant effect of the SFMIC treatments on the community structure, composition, and diversity. Earthworms' presence had been related to higher fungal diversity during lead phytoremediation [74]. In our study, the shifts caused by the SFMIC treatments were driven by *Chaetomium*, *Zopfiella* and *Sphaerosporella*. Here, *Chaetomium* was more abundant in the presence of collembolans, and, accordingly, several springtail species graze on *Chaetomium globosum* hyphae and disperse their spores [14, 75, 76]. *Chaetomium* also dominated during oxytetracycline degradation in greenhouse mesocosms inoculated with earthworms and arbuscular mycorrhizal fungi [77]. *Zopfiella* was more abundant in our pots containing collembolans and earthworms, in line with its dominance during vermicomposting [78]. In contrast, *Zopfiella* was less abundant in the absence of earthworms in a farmland soils [79]. Some *Zopfiella* species produce zopfiellin, a secondary metabolite with antifungal properties [80], that could be involved in soil pathogen suppression [81], and might have indirectly affected the fungal community in our experiment. Furthermore, collembolans feed preferably on pathogenic fungi than on mycorrhizal hyphae [82], which could explain the higher relative abundance of the ectomycorrhizal fungi *Sphaerosporella* in more complex SFMIC treatments.

Our second hypothesis was that the willows subjected to the highest level of SFMIC would be larger. All faunal treatments resulted in taller trees that produced more biomass as compared to the microbes-only contaminated pots, probably because phenanthrene decreased growth and soil fauna increased it. These results were, however, not always significant, and there was no difference between the different levels of SFMIC. Previous studies reported that

willows grow less under PAHs [83] or mixed contamination [84]. In contrast, and in line with our results, earthworms enhance plant productivity under natural conditions [85] and during phytoremediation [74]. This enhancement of plant growth by earthworms was linked to bioturbation (burrowing) that improves soil porosity [86]—leading to increased soil water infiltration [87] and aeration [15]—organic matter degradation, and nutrient cycling [88]. Bacterivorous nematodes, such as the one used in this study, also have the potential to stimulate plant growth through an increased mineralization of soil nutrients [89].

Finally, we hypothesized that willows growing in soils with a higher SFMIC would rhizodegrade more phenanthrene. Although we did find some effect of SFMIC on microbial communities and willow growth, which are the two most important factors for rhizoremediation efficiency, we did not find SFMIC-related differences in soil phenanthrene concentration after 8 weeks of growth. Accordingly, the PAH-degraders were, in general, not affected by the SFMIC treatments. As previously reported, PAH degraders were mostly affected by the contamination levels, and increased in abundance in the contaminated pots [26, 90]. These results are in sharp contradiction with previous studies that reported a positive direct effect of soil fauna, especially earthworms, on the degradation of different types of pollutants [70–72]. A recent study from our group showed that initial soil physicochemical and microbiological characteristics are critical for effective phytoremediation, with poplar trees only significantly affecting phenanthrene degradation in one of the two soils tested [22]. Similarly, it is possible that because of the initial characteristics of the soil used here, the effect of SFMIC on degradation rates was unnoticed. In fact, in most pots, both for rhizosphere and bulk soil, over 95% of the applied phenanthrene was degraded, suggesting that plant presence was not effective in enhancing degradation under our conditions. Alternatively, the unique sampling point after eight weeks of growth might have precluded the observation of any differences in degradation that might have occurred before that. Indeed, the addition of root exudates to phenanthrene contaminated sand resulted in significant reductions of the phenanthrene contamination compared to the unamended controls after only 10 days [91]. Other studies have shown a complete degradation of phenanthrene by willows within 3 months [92], 2 months [22] or as little as 21 days [93]. Additional studies using different SFMIC levels where the contaminant is evaluated at several time points throughout the experiment and in soils with different biological and physico-chemical characteristics will be needed to rule out if soil fauna influences the degradation rates during phytoremediation.

Even though most of the contaminant had disappeared, large differences in microbial communities, including PAH degraders abundance, diversity and community composition, were still visible between contaminated and control pots, suggesting some level of legacy effect of contamination. The phenanthrene contamination triggered a general reduction of the fungal diversity, which was driven by large increases in the relative abundance of *Sphaerosporella*. The predominance of this fungus has already been reported in other pot and field phytoremediation experiments carried out with trees [22, 27, 36, 94, 95]. Furthermore, its presence in the rhizosphere or in the bulk soil of pots with tree cuttings has been related to an enhancement of plant biomass [27] and to resistance to contamination stress [28, 36, 95], probably through its role as an ectomycorrhiza [95]. In our study, *Sphaerosporella* relative abundance was significantly higher in contaminated pots, supporting the premise that this fungus may be helping *Salicaceae* trees survive in stressful environments. Quite interestingly, *Sphaerosporella* relative abundance was also significantly affected by the SFMIC treatments, being generally higher in more complex treatments, which could contribute to explain the effects of SFMIC on plant growth parameters.

## Conclusion

Soils are heterogeneous ecosystems, harbouring rich communities, where meso- and macro-fauna shape physicochemical and microbiological characteristics [11, 14, 15, 18, 96]. Trophic interactions between fauna and microorganisms can shift microbial communities, affecting phytoremediation [39, 40]. However, there is a lack of experiments exploring the effect of these complex interactions. In this context, our unique experimental design allowed us to disentangle the effects of the various components of the fauna-microbial interactions complexity (SFMIC) on phytoremediation efficiency. Despite the effect of SFMIC on microbial communities, especially among fungi, we did not find shifts among PAH degraders. Accordingly, even though animals increased the willow's growth, phenanthrene degradation was unaffected by the SFMIC treatments. The phytoremediation results were not as expected from studies looking at the different components in isolation, highlighting the value of an approach encompassing the complex interactions occurring during phytoremediation. Incorporating meso- and macrofauna in controlled pot studies yields a better representation of field conditions, helping us understand phytoremediation. Future experiments with varying faunal biomass, and including multiple sampling times would help linking microbial community shifts to contaminant fate. Optimizing these interactions could lead to a wider adoption of green technologies to decontaminate polluted soils.

## Supporting information

**S1 Table A. Number of Miseq sequence reads per library.** B. Count reports for generated ASV.
(ZIP)

**S2 Table. Linear regression models, ANOVA and Tukey tests for testing the effect of SFMIC and contamination on plant traits, and to test the effect of compartment and SFMIC on residual phenanthrene.**
(XLSX)

**S3 Table. Median Shannon H' diversity and median ASV.**
(DOCX)

**S4 Table. Linear regression models, ANOVA and Tukey tests for testing the effect of factors on diversity indices.**
(XLSX)

**S5 Table. Linear regression models, ANOVA and Tukey tests for testing the effect of biomass and residual phenanthrene on diversity indices.**
(XLSX)

**S6 Table. ANOVA summary tables for the relative abundance of fungi at genus level.**
(DOCX)

**S7 Table. ANOVA summary tables for the relative abundance of bacteria at phylum level.**
(DOCX)

**S8 Table. ANOVA summary tables for the relative abundance of PAH-RHD degradation genes for the Gram-negative genetic marker, at the genus level.**
(DOCX)

**S9 Table. ANOVA summary tables for the relative abundance of PAH-RHD degradation genes for the Gram-positive genetic marker, at the genus level.**
(DOCX)

**S10 Table. Summary of the three-way analysis of the variance (ANOVA) on the absolute abundance of PAH-RHD genes (on copy numbers per gram of soil).**
(DOCX)

**S1 Fig. Relative abundance of individual bacterial phyla by soil food microbial interaction complexity treatment and compartment.**
(TIF)

## Acknowledgments

We thank the precious help provided by Denis Lachance during the setup of the experiment, as well as all the members of the Laurentian Forestry Centre and the Centre Armand Frappier Santé Biotechnologie who contributed with ideas and discussions to improve the outcome of this study. We also thank the logistic support provided by Stepan Pasharyan during the experimental set up.

## Author Contributions

**Conceptualization:** Sara Correa-Garcia, Armand Séguin, Etienne Yergeau.

**Data curation:** Sara Correa-Garcia, Jessica Ann Dozois, Eugenie Mukula.

**Formal analysis:** Sara Correa-Garcia, Julien Tremblay.

**Funding acquisition:** Armand Séguin, Etienne Yergeau.

**Investigation:** Sara Correa-Garcia, Vincenzo Corelli, Julien Tremblay, Armand Séguin, Etienne Yergeau.

**Methodology:** Sara Correa-Garcia, Vincenzo Corelli, Julien Tremblay, Jessica Ann Dozois, Eugenie Mukula, Etienne Yergeau.

**Project administration:** Etienne Yergeau.

**Resources:** Armand Séguin, Etienne Yergeau.

**Software:** Julien Tremblay.

**Supervision:** Armand Séguin, Etienne Yergeau.

**Validation:** Etienne Yergeau.

**Visualization:** Sara Correa-Garcia.

**Writing – original draft:** Sara Correa-Garcia, Etienne Yergeau.

**Writing – review & editing:** Sara Correa-Garcia, Vincenzo Corelli, Julien Tremblay, Jessica Ann Dozois, Armand Séguin, Etienne Yergeau.

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
