## [Decision Letter · Decision Letter 0]

7 Aug 2023

PONE-D-23-18792Soil fauna-microbial interactions complexity triggers shifts in microbial communities under a contamination disturbance.PLOS ONE

Dear Dr. Yergeau,

Thank you for submitting your manuscript to PLOS ONE. After careful consideration, we feel that it has merit but does not fully meet PLOS ONE’s publication criteria as it currently stands. Therefore, we invite you to submit a revised version of the manuscript that addresses the points raised during the review process.

Please submit your revised manuscript by Sep 21 2023 11:59PM. f you will need more time than this to complete your revisions, please reply to this message or contact the journal office at plosone@plos.org. Please include the following items when submitting your revised manuscript:A rebuttal letter that responds to each point raised by the academic editor and reviewer(s). You should upload this letter as a separate file labeled 'Response to Reviewers'.A marked-up copy of your manuscript that highlights changes made to the original version. You should upload this as a separate file labeled 'Revised Manuscript with Track Changes'.An unmarked version of your revised paper without tracked changes. You should upload this as a separate file labeled 'Manuscript'.If applicable, we recommend that you deposit your laboratory protocols in protocols.io to enhance the reproducibility of your results. Protocols.io assigns your protocol its own identifier (DOI) so that it can be cited independently in the future. For instructions see: https://journals.plos.org/plosone/s/submission-guidelines#loc-laboratory-protocols. Additionally, PLOS ONE offers an option for publishing peer-reviewed Lab Protocol articles, which describe protocols hosted on protocols.io. Read more information on sharing protocols at https://plos.org/protocols?utm_medium=editorial-email&utm_source=authorletters&utm_campaign=protocols.

We look forward to receiving your revised manuscript.

Kind regards,

Tunira Bhadauria, Ph.D.

Academic Editor

PLOS ONE

Journal Requirements:

"This work was supported by the Natural Sciences and Engineering Research Council of Canada (Discovery grant RGPIN-2014-05274 and strategic grant for projects STPGP 494702) to E.Y

S.C.G. was supported by the Research Affiliate Program from the Government of Canada. V.C. and J.A.D. were both supported by the Undergraduate Student Research Awards from the Natural Sciences and Engineering Research Council of Canada. E.M.K. was supported by a scholarship from the Armand-Frappier Foundation. This research was enabled in part by support provided by Calcul Québec (www.calculquebec.ca) and Compute Canada. "

"This work was supported by the Natural Sciences and Engineering Research Council of Canada (Discovery grant RGPIN-2014-05274 and strategic grant for projects STPGP 494702) to E.Y S.C.G. was supported by the Research Affiliate Program from the Government of Canada. V.C. and J.A.D. were both supported by the Undergraduate Student Research Awards from the Natural Sciences and Engineering Research Council of Canada. E.M.K. was supported by a scholarship from the Armand-Frappier Foundation. This research was enabled in part by support provided by Calcul Québec (www.calculquebec.ca) and Compute Canada."

"This work was supported by the Natural Sciences and Engineering Research Council of Canada (Discovery grant RGPIN-2014-05274 and strategic grant for projects STPGP 494702) to E.Y

S.C.G. was supported by the Research Affiliate Program from the Government of Canada. V.C. and J.A.D. were both supported by the Undergraduate Student Research Awards from the Natural Sciences and Engineering Research Council of Canada. E.M.K. was supported by a scholarship from the Armand-Frappier Foundation. This research was enabled in part by support provided by Calcul Québec (www.calculquebec.ca) and Compute Canada."

Reviewers' comments:

Reviewer's Responses to Questions

**Comments to the Author**

1. Is the manuscript technically sound, and do the data support the conclusions?

Reviewer #1: Yes

Reviewer #2: Yes

2. Has the statistical analysis been performed appropriately and rigorously? 

Reviewer #1: Yes

Reviewer #2: Yes

3. Have the authors made all data underlying the findings in their manuscript fully available?

Reviewer #1: Yes

Reviewer #2: Yes

4. Is the manuscript presented in an intelligible fashion and written in standard English?

Reviewer #1: Yes

Reviewer #2: Yes

5. Review Comments to the Author

Reviewer #1: Author are recommended to do needful

1.Introduction must be expanded

2.future prospective must be strengthen

3.An abstract figure or work flow must be prepared

4.RT PCR and Illumina Seq work are remarkable and highly appreciable

Reviewer #2: Dear Authors, The manuscript has been written very well and the research plan is very scientific and sound. The MS has shown the original findings and contradictory results with concerned previous studies. It is always good to show the all aspects of research findings either positive or negative.

However, it is suggested to revise the title of the manuscript as per finding if possible.

Line number 76, potting mix is okay or potting mixture?

Kindly rewrite the sentence , line number 188-189.

Use brackets to before writing supplementary table name and number in the MS. ex. for line number 201-202

Kindly check for all the abbreviations used that must be in full form at least ones in the MS.

The abstract section will be better if avoid too much abbreviations.

The conclusion section is too crispy. Therefore, authors can elaborate it in few more lines.

The community composition section in the results has been written vastly, authors can reduce these lines by citing concerned tables to increase the readability. For examples, all the species name is not essential to write, it can be referred to the tables.

The data analysis and statistical analysis part have been performed very well.

The MS can be accepted in this form after suggested few minor corrections.

6. PLOS authors have the option to publish the peer review history of their article (what does this mean?). If published, this will include your full peer review and any attached files.

Reviewer #1: **Yes: **Dr Kuldip Jayaswall

Reviewer #2: **Yes: **Deepanshu Jayaswal

---

## [Author Response · Author response to Decision Letter 0]

31 Aug 2023

Response to reviewers

Journal Requirements:

https://journals.plos.org/plosone/s/file?id=wjVg/PLOSOne_formatting_sample_main_body.pdf and  https://journals.plos.org/plosone/s/file?id=ba62/PLOSOne_formatting_sample_title_authors_affiliations.pdf

***We have now adapted the manuscript to the style requirements.

***We have now revised the Funding statement. 

"This work was supported by the Natural Sciences and Engineering Research Council of Canada (Discovery grant RGPIN-2014-05274 and strategic grant for projects STPGP 494702) to E.Y S.C.G. was supported by the Research Affiliate Program from the Government of Canada. V.C. and J.A.D. were both supported by the Undergraduate Student Research Awards from the Natural Sciences and Engineering Research Council of Canada. E.M.K. was supported by a scholarship from the Armand-Frappier Foundation. This research was enabled in part by support provided by Calcul Québec (www.calculquebec.ca) and Compute Canada. "

Please state what role the funders took in the study. If the funders had no role, please state: "The funders had no role in study design, data collection and analysis, decision to publish, or preparation of the manuscript."  If this statement is not correct you must amend it as needed. 

***We have now specified the role of the Funder.

"This work was supported by the Natural Sciences and Engineering Research Council of Canada (Discovery grant RGPIN-2014-05274 and strategic grant for projects STPGP 494702) to E.Y S.C.G. was supported by the Research Affiliate Program from the Government of Canada. V.C. and J.A.D. were both supported by the Undergraduate Student Research Awards from the Natural Sciences and Engineering Research Council of Canada. E.M.K. was supported by a scholarship from the Armand-Frappier Foundation. This research was enabled in part by support provided by Calcul Québec (www.calculquebec.ca) and Compute Canada."

"This work was supported by the Natural Sciences and Engineering Research Council of Canada (Discovery grant RGPIN-2014-05274 and strategic grant for projects STPGP 494702) to E.Y S.C.G. was supported by the Research Affiliate Program from the Government of Canada. V.C. and J.A.D. were both supported by the Undergraduate Student Research Awards from the Natural Sciences and Engineering Research Council of Canada. E.M.K. was supported by a scholarship from the Armand-Frappier Foundation. This research was enabled in part by support provided by Calcul Québec (www.calculquebec.ca) and Compute Canada."

***We have now removed the funding related text from the manuscript

***We have updated your Data Availability statement to reflect the information you provide in your cover letter.

We have now published the minimal underlying data in NCBI (to reproduce the analyses from raw sequences). We have also published the processed count matrices and the data related to the plant, quantitative PCR and the phenanthrene in Zenodo. https://doi.org/10.5281/zenodo.8299107

***Please, see comment above.

***We have now included captions for the Supporting information at the end of the manuscript.

8. Please review your reference list to ensure that it is complete and correct. If you have cited papers that have been retracted, please include the rationale for doing so in the manuscript text, or remove these references and replace them with relevant current references. Any changes to the reference list should be mentioned in the rebuttal letter that accompanies your revised manuscript. If you need to cite a retracted article, indicate the article’s retracted status in the References list and also include a citation and full reference for the retraction notice.     Reviewers' comments:  Reviewer's Responses to Questions

***We have updated the reference style and revised the reference list. 

Comments to the Author  1. Is the manuscript technically sound, and do the data support the conclusions?  The manuscript must describe a technically sound piece of scientific research with data that supports the conclusions. Experiments must have been conducted rigorously, with appropriate controls, replication, and sample sizes. The conclusions must be drawn appropriately based on the data presented.

Reviewer #1: Yes

Reviewer #2: Yes

2. Has the statistical analysis been performed appropriately and rigorously?

Reviewer #1: Yes

Reviewer #2: Yes

3. Have the authors made all data underlying the findings in their manuscript fully available?  The PLOS Data policy requires authors to make all data underlying the findings described in their manuscript fully available without restriction, with rare exception (please refer to the Data Availability Statement in the manuscript PDF file). The data should be provided as part of the manuscript or its supporting information, or deposited to a public repository. For example, in addition to summary statistics, the data points behind means, medians and variance measures should be available. If there are restrictions on publicly sharing data—e.g. participant privacy or use of data from a third party—those must be specified.

Reviewer #1: Yes

Reviewer #2: Yes

4. Is the manuscript presented in an intelligible fashion and written in standard English?  PLOS ONE does not copyedit accepted manuscripts, so the language in submitted articles must be clear, correct, and unambiguous. Any typographical or grammatical errors should be corrected at revision, so please note any specific errors here.

Reviewer #1: Yes

Reviewer #2: Yes

5. Review Comments to the Author  Please use the space provided to explain your answers to the questions above. You may also include additional comments for the author, including concerns about dual publication, research ethics, or publication ethics. (Please upload your review as an attachment if it exceeds 20,000 characters)

Reviewer #1: 

Author are recommended to do needful 

1.Introduction must be expanded 

***Done, we have now added a section at the beginning of the introduction where we explore in depth the PAH problematic and phytoremediation as a bioremediation approach suitable for PAH contamination.

2.future prospective must be strengthen 

***Done. We have now provided both context and future perspectives to the conclusion section.

3.An abstract figure or work flow must be prepared 

***Done. We have now provided an abstract figure to illustrate the experimental design.

4.RT PCR and Illumina Seq work are remarkable and highly appreciable

***Thank you very much!

Reviewer #2: 

Dear Authors, The manuscript has been written very well and the research plan is very scientific and sound. The MS has shown the original findings and contradictory results with concerned previous studies. It is always good to show the all aspects of research findings either positive or negative. 

***Thank you very much!

However, it is suggested to revise the title of the manuscript as per finding if possible. 

***We have now updated the title to be more explicit with the results. 

Line number 76, potting mix is okay or potting mixture? 

***Thank you, we have now changed this (L102, 110, 134).

Kindly rewrite the sentence, line number 188-189. 

***Done, L217 now read as: “Standards were made from 10-fold dilutions of linearized plasmid containing the gene fragment of interest, cloned from soil DNA”

Use brackets to before writing supplementary table name and number in the MS. ex. for line number 201-202 

***Thank you, we have now corrected this throughout the text to conform to PLOS ONE style.

Kindly check for all the abbreviations used that must be in full form at least ones in the MS. 

***Done, we have now spelled out all abbreviations on their first mention throughout the manuscript, especially in the material and methods section. 

The abstract section will be better if avoid too much abbreviations. 

***We have now reduced the abbreviations in the abstract.

The conclusion section is too crispy. Therefore, authors can elaborate it in few more lines. 

***We have now elaborated this section, both in context and perspectives.

The community composition section in the results has been written vastly, authors can reduce these lines by citing concerned tables to increase the readability. For examples, all the species name is not essential to write, it can be referred to the tables. 

***We have now reduced this part of the results section.

The data analysis and statistical analysis part have been performed very well. The MS can be accepted in this form after suggested few minor corrections.

***Thank you very much for your kind comments!

6. PLOS authors have the option to publish the peer review history of their article (what does this mean?). If published, this will include your full peer review and any attached files.   Do you want your identity to be public for this peer review? For information about this choice, including consent withdrawal, please see our Privacy Policy.

Reviewer #1: Yes: Dr Kuldip Jayaswall

Reviewer #2: Yes: Deepanshu Jayaswal

---

## [Editor Report · Decision Letter 1]

18 Sep 2023

Soil fauna-microbial interactions shifts fungal and bacterial communities under a contamination disturbance.

PONE-D-23-18792R1

Dear Dr. Yergeau

We’re pleased to inform you that your manuscript has been judged scientifically suitable for publication and will be formally accepted for publication once it meets all outstanding technical requirements.

Kind regards,

Tunira Bhadauria, Ph.D.

Academic Editor

PLOS ONE
---

## [Editor Report · Acceptance letter]

13 Oct 2023

PONE-D-23-18792R1 

Soil fauna-microbial interactions shifts fungal and bacterial communities under a contamination disturbance. 

Dear Dr. Yergeau:

I'm pleased to inform you that your manuscript has been deemed suitable for publication in PLOS ONE. Congratulations! Your manuscript is now with our production department. 

Kind regards, 

on behalf of

Dr. Tunira Bhadauria 

Academic Editor

PLOS ONE